# Eckol Inhibits Particulate Matter 2.5-Induced Skin Keratinocyte Damage via MAPK Signaling Pathway

**DOI:** 10.3390/md17080444

**Published:** 2019-07-27

**Authors:** Ao Xuan Zhen, Yu Jae Hyun, Mei Jing Piao, Pincha Devage Sameera Madushan Fernando, Kyoung Ah Kang, Mee Jung Ahn, Joo Mi Yi, Hee Kyoung Kang, Young Sang Koh, Nam Ho Lee, Jin Won Hyun

**Affiliations:** 1School of Medicine, Jeju National University, Jeju 63243, Korea; 2Laboratory of Veterinary Anatomy, College of Veterinary Medicine, Jeju National University, Jeju 63243, Korea; 3Department of Microbiology and Immunology, College of Medicine, Inje University, Busan 47392, Korea; 4Department of Chemistry and Cosmetics, College of Natural Sciences, Jeju National University, Jeju 63243, Korea

**Keywords:** phlorotannin, particulate matter, reactive oxygen species, keratinocytes

## Abstract

Toxicity of particulate matter (PM) towards the epidermis has been well established in many epidemiological studies. It is manifested in cancer, aging, and skin damage. In this study, we aimed to show the mechanism underlying the protective effects of eckol, a phlorotannin isolated from brown seaweed, on human HaCaT keratinocytes against PM_2.5_-induced cell damage. First, to elucidate the underlying mechanism of toxicity of PM_2.5_, we checked the reactive oxygen species (ROS) level, which contributed significantly to cell damage. Experimental data indicate that excessive ROS caused damage to lipids, proteins, and DNA and induced mitochondrial dysfunction. Furthermore, eckol (30 μM) decreased ROS generation, ensuring the stability of molecules, and maintaining a steady mitochondrial state. The western blot analysis showed that PM_2.5_ promoted apoptosis-related protein levels and activated MAPK signaling pathway, whereas eckol protected cells from apoptosis by inhibiting MAPK signaling pathway. This was further reinforced by detailed investigations using MAPK inhibitors. Thus, our results demonstrated that inhibition of PM_2.5_-induced cell apoptosis by eckol was through MAPK signaling pathway. In conclusion, eckol could protect skin HaCaT cells from PM_2.5_-induced apoptosis via inhibiting ROS generation.

## 1. Introduction

Natural compounds can be effective candidates for various skin diseases. Particularly, phlorotannins extracted from seaweeds have interesting properties that make them useful for cosmeceutical applications. They can whiten the skin by inhibiting melanin synthesis [1], and delay skin wrinkles by inhibiting matrix metalloproteinase [2]. Moreover, phlorotannins show antioxidant [3], anti-inflammatory [4], and hair-growth promotion activities [5]. Studies have shown that eckol, which is a kind of phlorotannin present in brown seaweeds (Phaeophyceae), decreases ultraviolet B (UVB)-induced oxidative stress in human keratinocytes at a dose of 27 μM [6], and inhibits cancer in SKH-1 mice via inhibiting UVB-induced inflammation [7], and declines matrix metalloproteinase-1 level in human dermal fibroblasts implying anti-aging effects at a dose of 10 μM [8]. Our earlier studies have proved that eckol could clear excess reactive oxygen species (ROS) and protect skin keratinocytes from apoptosis [6]. 

Air pollution by continuous emission of various pollutants into the atmosphere has led to the rapid decline of public health and accelerated climate change [9]. In addition, more than 90% population in the world breathed unhealthy air in 2017, which suggests that particulate pollution is a global challenge [10]. According to previous studies, particulate matter (PM) increases the public health risks for various diseases, such as respiratory disease [11], cardiovascular disease [12], and lung inflammation [13]. Fine particulate matter with a diameter less than 2.5 μm, denoted as PM_2.5,_ is present in the air for over several hours to weeks [14]. Significantly, PM_2.5_ could deeply penetrate the skin and the respiratory tract [15]. Skin damage caused by PM_2.5_ is manifested as inflammatory skin diseases, such as atopic dermatitis, acne, and psoriasis, aging, and cancer via multiple signaling pathways [16].

A recent study presents a comprehensive summary of the characteristics of eckol relevant for its therapeutic potential, including antioxidant activity following exposure to H_2_O_2_, radiation, and PM [17]. PM_2.5_ is a fine particulate matter that causes skin apoptosis related to the ROS generation [16], and it would be interesting to investigate whether eckol protected skin cells from ROS-induced damage. Moreover, the mode of action of eckol on PM_2.5_ is not well-established. Here, we have investigated the potential benefits of eckol on keratinocytes by studying its inhibitory effect on molecular damage, mitochondrial dysfunction, apoptosis-related factors, and MAPK signaling related proteins. In this study, our aim was also to gain insights into the mechanism underlying the protective action of eckol on PM_2.5_-induced skin cell apoptosis. 

## 2. Results

### 2.1. Eckol Showed Anti-oxidative Effects to Protect Cells from PM_2.5_-Induced Apoptotic Cell Death

Previous studies have shown that eckol exhibited no cytotoxicity to HaCaT cells at a concentration of 30 μM [18] but showed antioxidant activity [6]. Therefore, in this study, we used 30 μM of eckol (Figure 1a) as the optimal concentration. From Figure 1b,c, it is evident that while PM_2.5_ increased the levels of ROS as indicated by 2’,7’-dichlorofluorescein diacetate (DCF-DA) staining, eckol inhibited intracellular ROS generation. The results demonstrated that PM_2.5_-induced ROS could accelerate cell apoptosis and death. To confirm that eckol could help cells escape from this damage, we checked nuclei integrity, and cell viability. According to results, PM_2.5_ treatment led to sub-G_1_ cell population after 24 h (Figure 1d), fragmented nuclei (Figure 1e), and cell death (Figure 1f). However, it was noted that following treatment with eckol, the apoptotic cell death was decreased, and the cell viability was also enhanced.

### 2.2. Eckol Protected Cells against PM_2.5_-Induced Intracellular Molecular Damage

Previous studies have shown that increment in ROS disrupted intracellular molecules involved in apoptosis [19,20]. Thus, we detected lipid peroxidation, protein carbonylation, and DNA damage. The confocal images show that PM_2.5_ caused generation of phosphine oxide, which is a marker of lipid peroxidation. However, this was reversed by treatment with eckol (Figure 2a). Moreover, PM_2.5_ aggravated protein carbonylation level, which was decreased by eckol treatment (Figure 2b). DNA lesions and strand breaks were studied by staining the cells with avidin-tetramethylrhodamine isothiocyanate (TRITC) conjugate (Figure 2c) and comet assay (Figure 2d). The data show that eckol guarded DNA against PM_2.5_. 

### 2.3. Eckol Prevented PM_2.5_-Induced Mitochondrial Dysfunction

Mitochondria play an important role in cellular energy production, and their biogenesis is related to synthesis of molecules, such as lipids and proteins, DNA transcription, and even cell apoptosis [21]. Next, we examined mitochondrial functions. Dihydrorhodamine 123 (DHR123) staining images show that mitochondrial ROS was accumulated in PM_2.5_-treated group. Whereas, ROS level was decreased by pretreatment with eckol (Figure 3a). Both flow cytometry (Figure 3b) and confocal microscopy (Figure 3c) data demonstrate that PM_2.5_ caused mitochondrial depolarization, which was arrested by treatment with eckol. Furthermore, the flux of mitochondrial calcium was increased in the PM_2.5_-treatment group, and it was decreased in eckol-treatment group, which was monitored using the calcium indicator, Rhod-2 acetoxymethyl ester (Rhod-2 AM), by confocal microscopy (Figure 3d) and flow cytometry (Figure 3e). 

### 2.4. Eckol Modulated PM_2.5_-Induced Apoptotic Factors

It has been reported that urban particulate pollution penetrates the skin barrier and causes apoptosis in keratinocytes by activating caspase-3 [22]. Therefore, we evaluated the levels of the proapoptotic protein-Bax, antiapoptotic protein-Bcl-2, and cleaved caspase-3 (Figure 4a). The protein levels of Bax and activated caspase-3 were increased by PM_2.5_, but expression of Bcl-2 was decreased by treatment with PM_2.5_. However, these were reversed by eckol treatment. To investigate whether PM_2.5_ could induce apoptosis, we counted apoptotic bodies via Hoechst 33342 dye staining (Figure 4b). The number of apoptotic cells in PM_2.5_ group surged four times compared to that in the control group. However, both eckol and Z-VAD-FMK (the caspase inhibitor) halted the apoptotic bodies induced by PM_2.5_. 

### 2.5. Eckol Reduced MAPK Signaling Pathway Activated by PM_2.5_

In a review, Sun et al. have pointed out that many anti-cancer therapeutics induced apoptosis by modulating the MAPK/ERK signaling pathway [23]. Thus, we checked the expression levels of MAPK-related proteins, ERK, p38, and JNK, and the results showed that PM_2.5_ could stimulate ERK, p38, and JNK (Figure 5a). However, eckol inhibited the activation of ERK, p38, and JNK. Next, we examined PM_2.5_-induced apoptotic bodies by treatment with MAPK pathway inhibitors, U0126, SB203580, and SP600125 (inhibitors of ERK, p38, and JNK, respectively), and the results showed that all these three inhibitors could reduce the number of apoptotic bodies (Figure 5b). In addition, eckol enhanced the anti-apoptotic effect of MAPK-related inhibitors. 

## 3. Discussion

There have been several investigations into the bioactivities of eckol, since it was first isolated from *Ecklonia cava* [3]. Eckol has multi-protective effects towards several cell lines, including lung fibroblast cells [24], human dermal fibroblasts [8], Chang liver cells [25], and human keratinocytes [6]. Furthermore, eckol is a compound with therapeutic potential in many areas, such as anti-oxidative stress [24], radioprotective action [26], antithrombotic and profibrinolytic activities [27], and anticancer activity [28]. Piao et al. studied PM_2.5_-induced ROS generation at different concentrations (25, 50, 75, and 100 µg/mL) for 24 h in HaCaT keratinocytes, and found that PM_2.5_ 50 µg/mL caused excessive ROS and skin dysfunction [29]. Usually, oxidative stress is caused by excessive accumulation of ROS or lack of the ability to eliminate them. PM_2.5_ produces large amounts of ROS beyond the clearance ability of cells [30]. In our study, eckol showed its ability to protect cells against PM_2.5_-induced ROS, cell cycle arrest, and apoptosis, and improved cell viability.

To explore the mechanism in detail, we checked the state of intracellular molecules such as lipids, protein, and DNA, which play various important roles in the cells [31]. Furthermore, intracellular macromolecular damage can be recognized as oxidative stress [32]. Our results demonstrated that PM_2.5_ indeed induced oxidation of molecules, whereas eckol relieved intracellular molecular damage. The review also points out that mitochondria-dependent ROS generation subsequently caused cell cycle arrest and apoptosis, which is ROS-mediated apoptosis via mitochondrial mechanism [33]. In addition, our previous studies showed that calcium level and mitochondrial membrane potential affect the function of mitochondria [30,34]. The data in Figure 3 show that PM_2.5_ increased the calcium level and depolarized mitochondrial membrane potential as compared to the control cells, whereas eckol regulated the mitochondria and maintained a stable state. The mechanism of mitochondrial damage is related to Bcl-2 proteins, which maintains mitochondrial membrane integrity [35]. The interaction between Bcl-2 and Bax also influences antiapoptosis [36]. Bcl-2 plays an anti-apoptotic role, whereas Bax is proapoptotic [37]. There is a complex crosslink between Bcl-2 family proteins and caspase proteins in cell apoptosis, in which Bcl-2 indirectly activates the caspase cascade [38]. The caspase-3 results in apoptosis induced by both extrinsic and intrinsic stimulus [39]. The results elucidated that except for the decrease of Bcl-2, Bax and cleaved caspase-3 (activated caspase-3) were increased by PM_2.5_. However, eckol reversed these effects. Then, we treated cells with caspase inhibitor (Z-VAD-FMK) and found that upon pretreatment with caspase inhibitor, the apoptotic bodies were decreased significantly. These data prove that caspase proteins contributed to cell apoptosis induced by PM_2.5_. MAPK signaling pathway plays a role in many systems of cell proliferation, migration, and apoptosis [23]. Many drugs are used to modulate MAPK signaling pathway to induce cell apoptosis in cells, such as lung cancer [40], human colorectal cancer [41], and cervical cancer HeLa cells [42]. Finally, we checked MAPK signaling pathway-related proteins, ERK, p38, and JNK. The results show that PM_2.5_ activated all three proteins, but eckol exhibited the ability to inactivate them. When we used inhibitors of ERK, p38, and JNK to treat PM_2.5_-damaged cells, the numbers of apoptotic bodies were decreased, similar to eckol. These data further prove that MAPK signaling pathway plays a vital role in the inhibition of PM_2.5_-induced apoptosis by eckol.

Although the protective effects of eckol on human keratinocytes from PM_2.5_-induced skin damage has been shown, there are limitations to this study. These results from in vitro experiments need to be validated by animal studies and clinical trials. Moreover, the concentration of air pollutants in the natural environment is different from the PM_2.5_ purchased from the company, which provide certain ingredients for reference. In the future, there should be in vivo animal trials on skin protection to elucidate the protective effects and side effects of eckol under the complicated living environments.

## 4. Materials and Methods 

### 4.1. Eckol and PM_2.5_

Eckol was provided by Professor Nam Ho Lee of Jeju National University (Jeju, Korea), which belonged to Phaeophyceae. Preparation of the extract and its purification was following the reported protocol [43]. The dried brown alga *Ecklonia cava* was extracted with 80% methanol and the crude extract was purified by HPLC. After purification, 20 mg of pure eckol was obtained from 1 kg dry weight of the brown algae. A stock solution of eckol was prepared by dimethyl sulfoxide (DMSO). The NIST particulate matter SRM 1650b (PM_2.5_) was bought from Sigma-Aldrich (St. Louis, MO, USA) and a stock solution in DMSO was prepared to obtain a concentration of PM_2.5_ at 25 mg/mL. DMSO was as the control.

### 4.2. Cell Culture

The human HaCaT keratinocytes were purchased from Cell Lines Service (Heidelberg, Germany) and were grown in Dulbecco’s modified Eagle’s medium (Life Technologies Co., Grand Island, NY, USA) with 10% heat-inactivated fetal calf serum, streptomycin (100 μg/mL), and penicillin (100 units/mL). The cells were cultured at 37 °C in an incubator in an atmosphere containing 5% CO_2_ [6]. 

### 4.3. ROS Detection

To examine the intracellular ROS scavenging ability of eckol, DCF-DA (Sigma-Aldrich) staining assay was performed. Cells (1.5 × 10^5^ cells/mL) were treated with eckol (30 µM) for 30 min, PM_2.5_ (50 µg/mL) for another 24 h, and DCF-DA (25 µM) sequentially. Then, the stained cells were detected by using a flow cytometer (Becton Dickinson, Franklin Lakes, NJ, USA) and confocal microscope (Carl Zeiss, Oberkochen, Germany). Similarly, mitochondrial ROS levels were detected by DHR123 (Molecular Probes) staining (10 μM) [30].

### 4.4. Sub-G_1_ Cell Detection

Cells were treated with eckol (30 µM) and PM_2.5_ (50 µg/mL) sequentially. After 24 h, cells were harvested and dyed with PI and RNase A (1:1000) for 30 min. Finally, the fluorescence emission was detected with a FACSCalibur flow cytometer (Becton Dickinson) [30]. 

### 4.5. Hoechst 33342 Staining

The apoptotic bodies were examined with Hoechst 33342 (Sigma-Aldrich), which is a nucleus-specific dye. All cells were stained with Hoechst 33342, and the images were captured under a Cool SNAP-Pro color digital camera (Media Cybernetics, Silver Spring, MD, USA) in a fluorescence microscope [44]. 

### 4.6. Cell Viability

Cells were cultured with eckol and/or PM_2.5_ for 24 h, and the dead cells were stained with 0.1% trypan blue solution. Then, unstained bodies (live cells) and stained bodies (dead cells) were counted separately. Cell viability (%) was determined as: Live cells/ (live cells + dead cells) × 100% [45].

### 4.7. Lipid Peroxidation Assay

Cells (1.5 × 10^5^ cells/mL) were cultured with eckol and/or PM_2.5_ for 24 h in the chamber slides. The lipid hydroperoxides in cells were reacted with diphenylpyrenylphosphine (DPPP, Molecular Probes) and the lipid adducts of DPPP oxide were detected by a confocal microscope [30].

### 4.8. Protein Carbonylation Assay

Cells (1.5 × 10^5^ cells/mL) were cultured with eckol and/or PM_2.5_ for 24 h in the culture dish. The Oxiselect^TM^ protein carbonyl ELISA kit (Cell Biolabs, San Diego, CA, USA) was used to detect protein oxidation following the manufacturer’s instructions [34].

### 4.9. Detection of 8-Oxoguanine (8-oxoG)

Cells (1.5 × 10^5^ cells/mL) were cultured with eckol and/or PM_2.5_ for 24 h in the chamber slides. The avidin-TRITC conjugate was used to detect 8-oxoG, an indicator of oxidative DNA damage. The stained cells were visualized under a confocal microscope [27].

### 4.10. Single Cell Gel Electrophoresis (Comet Assay)

Cells (1.5 × 10^5^ cells/mL) were cultured with eckol and/or PM_2.5_ for 30 min in the microtubes, and the harvested cells were fixed on a microscopic slide with low-melting agarose (1%). The slides with cells were permeated into lysis buffer (pH 10) for 1 h, which contained NaCl (2.5 M), Na-EDTA (100 mM), Tris (10 mM), Triton X-100 (1%), and DMSO (10%). Finally, the slides were subjected to electrophoresis, and the samples were dyed with ethidium bromide. The fluorescent images of the tails were captured with a fluorescence microscope and the tail lengths (50 cells per slide) were analyzed by the image analysis software (Kinetic Imaging, Komet 5.5, Liverpool, UK) [29].

### 4.11. Quantification of Ca^2+^ Level

Cells (1.5 × 10^5^ cells/mL) were cultured with eckol and/or PM_2.5_ for 24 h in the chamber slides. Cells were stained with Rhod-2 AM to check mitochondrial calcium levels, and the images were captured by confocal microscopy and flow cytometry [30].

### 4.12. Mitochondrial Membrane Potential (ΔΨm) Analysis

Cells (1.5 × 10^5^ cells/mL) were cultured with eckol and/or PM_2.5_ for 24 h in the chamber slides. Cells were stained with 5,5’,6,6’-tetrachloro-1,1’,3,3’-tetraethylbenzimidazolylcarbocyanine iodide (JC-1, Invitrogen, Carlsbad, CA, USA) to observe changes in cell membrane potential. The fluorescence emission was analyzed by confocal microscopy and flow cytometry [6].

### 4.13. Western Blotting 

The protein lysates from harvested cells were subjected to SDS-PAGE and transferred into membranes in sequence. The membranes were incubated with primary and secondary antibody (Pierce, Rockford, IL, USA) for 1 h separately. Protein bands were visualized via X-ray film (AGFA, Belgium). The following primary antibodies were used: caspase-3, phospho-p38, phospho-ERK, and phospho-JNK (Cell Signaling Technology, Beverly, MA, USA), Bax and Bcl-2 (Santa Cruz Biotechnology, Santa Cruz, CA, USA), and actin (Sigma-Aldrich) [44]. 

### 4.14. Statistical Analysis

All data were collected from three experiments and presented as mean ± standard error, which were analyzed by variance (ANOVA) with Tukey’s test using Sigma Stat (v12) software (SPSS, Chicago, IL, USA). *P*-values < 0.05 were considered statistically significant.

## 5. Conclusions

PM_2.5_ causes skin cell damage by generating ROS, excess of which oxidizes intracellular molecules and causes mitochondrial dysfunction, and activates MAPK signaling pathways. The skin is the first protection from various pollutants in the air, and it is important to identify effective biological compounds to prevent skin damage. Eckol has been known to inhibit UVB-induced ROS generation in keratinocytes [6]. In this study, we show that eckol could protect keratinocytes from PM_2.5_-induced apoptosis by halting ROS generation, suggesting that eckol is a suitable candidate for skin protection and can be useful in the cosmetics industry as well as in medicine.

## Figures and Tables

**Figure 1 marinedrugs-17-00444-f001:**
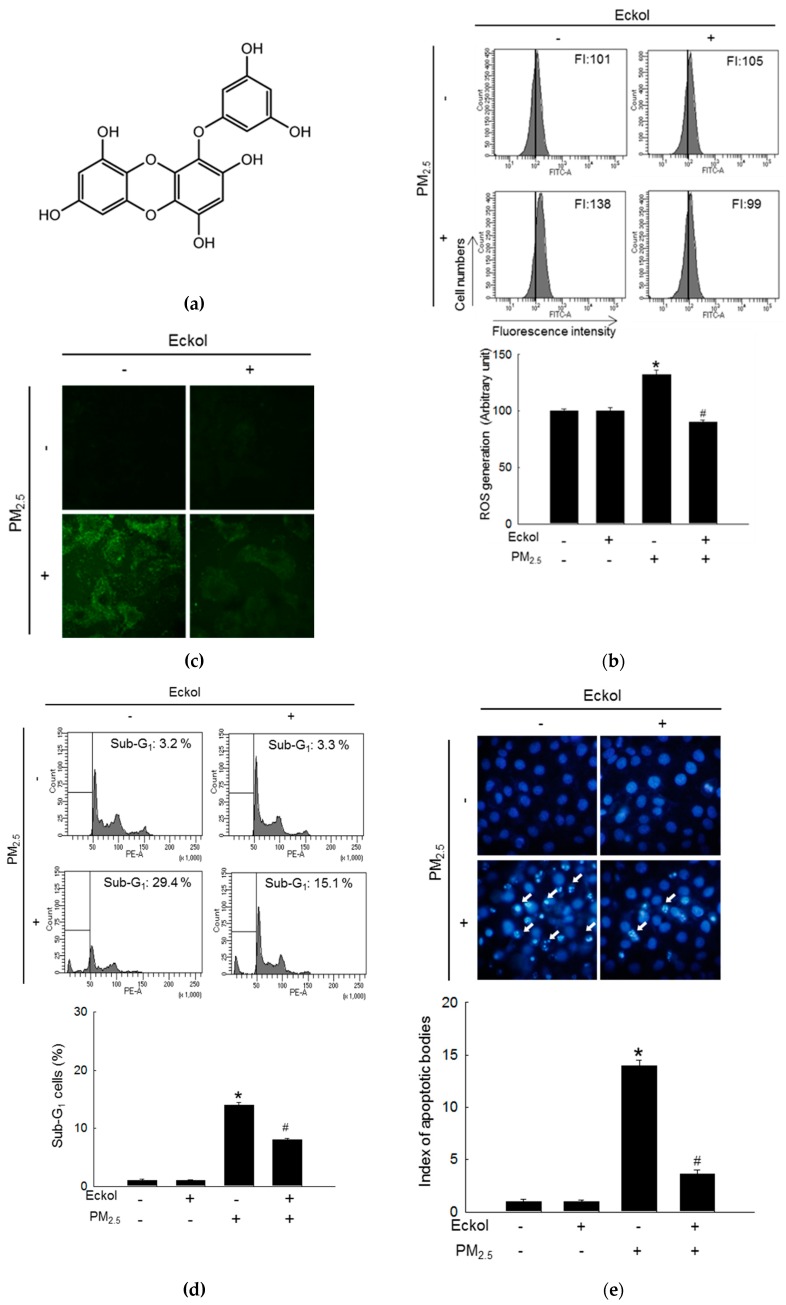
Eckol (30 µM) decreased cell apoptotic bodies by inhibiting PM_2.5_-induced ROS level. (**a**) Chemical structure of eckol. Intracellular ROS level (DCF-DA staining) induced by PM_2.5_ (50 µg/mL) was inhibited via treatment with eckol as observed by (**b**) flow cytometry and (**c**) confocal. (**d**) Sub-G_1_ cell population induced by PM_2.5_ was blocked by treatment with eckol, as determined by propidium iodide staining. (**e**) Apoptosis induced by PM_2.5_ was reduced by treatment with eckol, observed by Hoechst 33342 staining. The arrow indicated the apoptotic bodies. (**f**) Cell deaths induced by PM_2.5_ were reduced via treatment with eckol, determined by trypan blue assay. The arrow indicated the dead cell (stained cells by trypan blue). All experiments were performed after treatment with PM_2.5_ for 24 h, and *n* = 3 for every group. * *p* < 0.05 and ^#^
*p* < 0.05 compared to control cells and PM_2.5_-exposed cells, respectively.

**Figure 2 marinedrugs-17-00444-f002:**
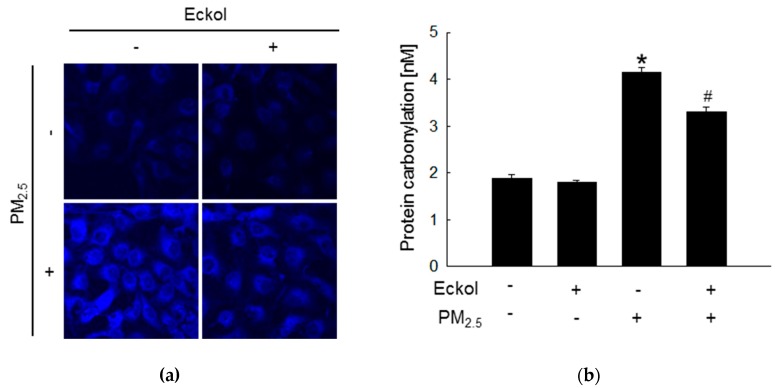
Eckol (30 µM) protected intracellular molecules from PM_2.5_-induced damage. (**a**) Lipid oxidation induced by PM_2.5_ was mitigated via treatment with eckol through diphenylpyrenylphosphine (DPPP) staining. (**b**) Protein carbonylation induced by PM_2.5_ was declined via treatment with eckol as observed by a protein carbonylation assay. DNA damage induced by PM_2.5_ was inhibited via treatment with eckol, as confirmed through (**c**) avidin-TRITC staining and (**d**) comet assay. All experiments were performed after treatment with PM_2.5_ for 24 h, and n = 3 for every group. * *p* < 0.05 and # *p* < 0.05 compared to control cells and PM_2.5_-exposed cells, respectively.

**Figure 3 marinedrugs-17-00444-f003:**
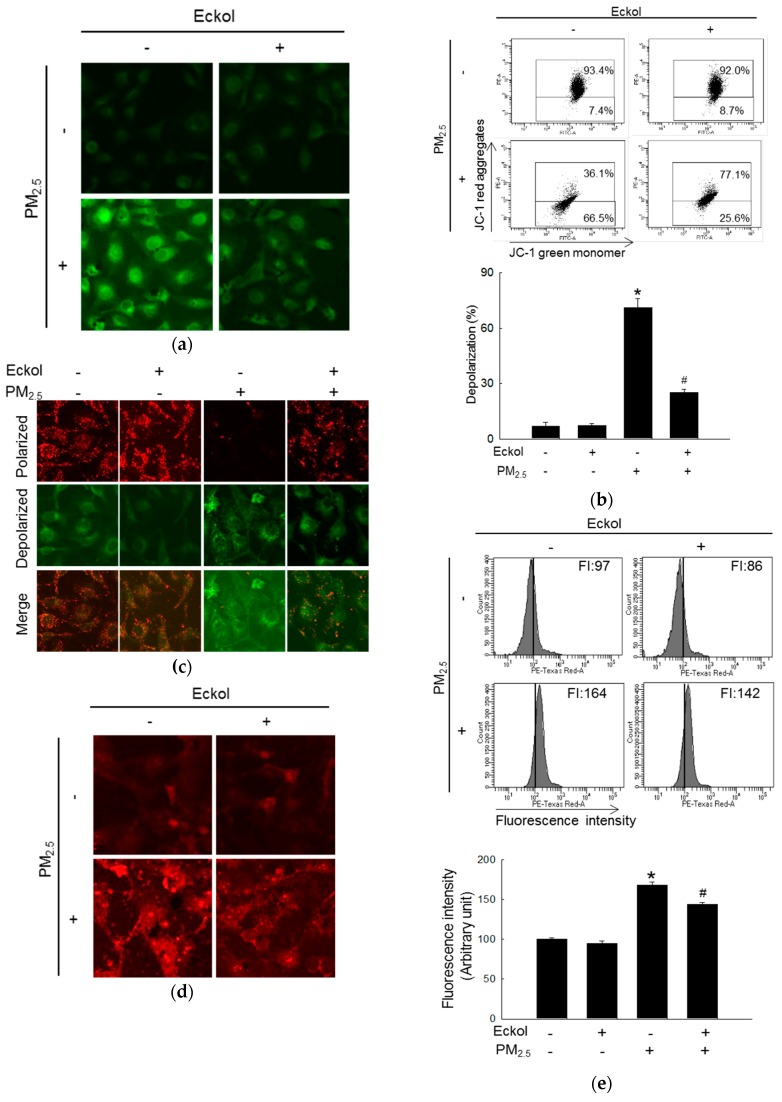
Eckol (30 µM) prevented PM_2.5_-induced mitochondrial dysfunction by balancing mitochondrial membrane potential and calcium level. (**a**) Mitochondrial ROS induced by PM_2.5_ was decreased via treatment with eckol through DHR123 staining. Depolarization of mitochondrial membrane potential (JC-1 staining) induced by PM_2.5_ was repolarized via treatment with eckol through (**b**) flow cytometry and (**c**) confocal microscopy. Extra-mitochondrial Ca^2+^ (Rhod-2 AM staining) induced by PM_2.5_ was blocked by treatment with eckol was monitored using (**d**) confocal microscopy and (**e**) flow cytometry. All experiments were performed after treatment with PM_2.5_ for 24 h, and *n* = 3 for every group. * *p* < 0.05 and ^#^
*p* < 0.05 compared to control cells and PM_2.5_-exposed cells, respectively.

**Figure 4 marinedrugs-17-00444-f004:**
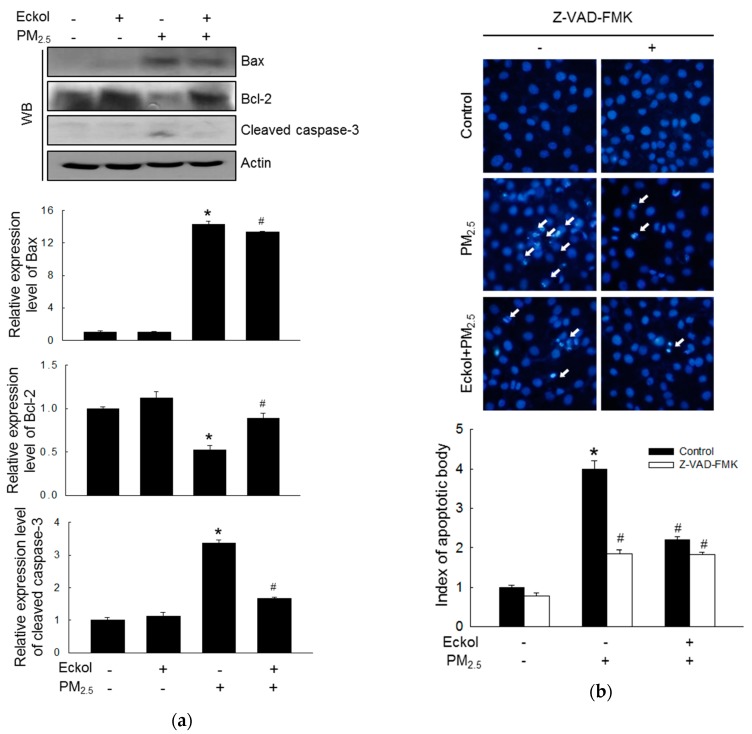
Eckol (30 µM) regulated apoptosis-related proteins induced by PM_2.5_. (**a**) Increase of Bax and cleaved caspase-3 and decrease of Bcl-2 by PM_2.5_ were reversed by treatment with eckol, as observed by western blotting (WB). (**b**) Apoptosis induced by PM_2.5_ was reduced by treatment with eckol or caspase inhibitor (Z-VAD-FMK), as seen by Hoechst 33342 staining. All experiments were performed after treatment with PM_2.5_ for 24 h, and *n* = 3 for every group. * *p* < 0.05 and ^#^
*p* < 0.05 compared to control cells and PM_2.5_-exposed cells, respectively.

**Figure 5 marinedrugs-17-00444-f005:**
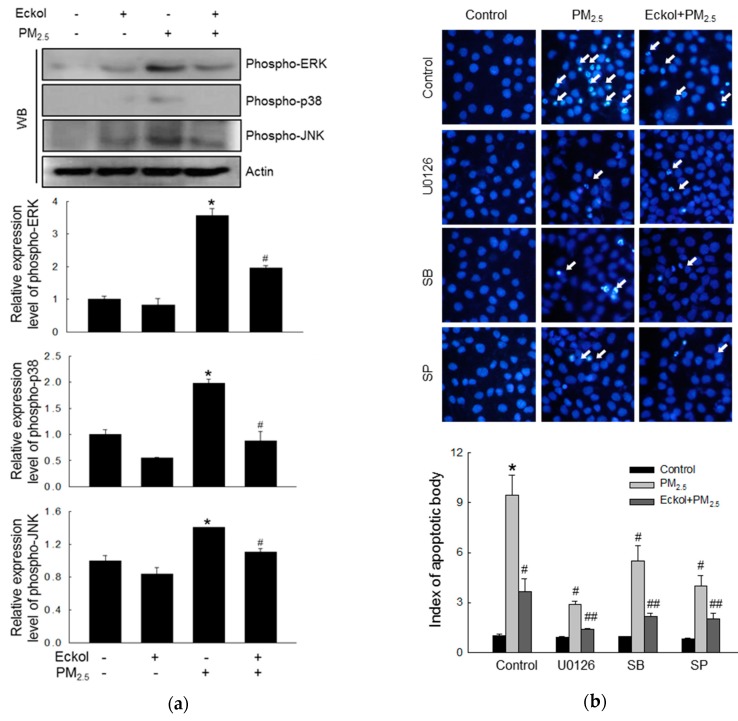
Eckol (30 µM) reduced PM_2.5_-induced MAPK signaling pathway. (**a**) Western blot showing that activation of ERK, p38, and JNK induced by PM_2.5_ was reversed via treatment with eckol. (**b**) Apoptosis induced by PM_2.5_ was reduced by treatment with eckol or ERK, p38, and JNK inhibitors (U0126, SB203580 (SB), and SP600125 (SP), respectively), as observed through Hoechst 33342 staining. All experiments were performed after treatment with PM_2.5_ for 24 h, and *n* = 3 for every group. * *p* < 0.05, ^#^
*p* < 0.05 and ^##^
*p* < 0.05 compared to control cells, PM_2.5_-exposed cells, and both inhibitor and PM_2.5_-exposed cells respectively.

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
