# Peer review of "Eckol Inhibits Particulate Matter 2.5-Induced Skin Keratinocyte Damage via MAPK Signaling Pathway"

_marinedrugs, 2019, doi:10.3390/md17080444_

Round 1

Reviewer 1 Report

Authors presented Eckol prevent PM2.5 induced apoptosis by MAPK signal,

The manuscript is very clear. I don't have too many concerns for this manuscript. 

Only a few minor concerns as below,

Figure 1,2,3,4,5 +/- signs could be larger for easy visualization.

Figure 1, flow data, x-axis legend were removed/missing. Please enlarge 1-e since I don't understand arrows mean.

Figure 2,3, please enhance blue, red and green signals. It is very difficult to see.

Figure 2, enlarge comet images

Hoechst33,342 or 33342? Please use one name.

Please explain in detail for materials and methods 4.7, 4.8, 4.9. 4.11, 4.12.

Author Response

Reviewer comment 1.

Authors presented Eckol prevent PM2.5 induced apoptosis by MAPK signal. The manuscript is very clear. I don't have too many concerns for this manuscript. 

Only a few minor concerns as below,

Figure 1,2,3,4,5 +/- signs could be larger for easy visualization.

Response: According to reviewer’s comment, we have enlarged the size of the figures and the symbols.

Figure 1, flow data, x-axis legend were removed/missing. Please enlarge 1-e since I don't understand arrows mean.

Response: According to reviewer’s comment, we have enlarged the size of the image in Figure 1e. The x-axis in the plot represents FITC-A.

Figure 2,3, please enhance blue, red and green signals. It is very difficult to see.

Figure 2, enlarge comet images

Response: According to reviewer’s comment, we have enhanced the blue, red, and green signals, and enlarged the size of the comet images.

Hoechst33,342 or 33342? Please use one name.

Response: It was meant to be Hoechst 33342. We have edited the manuscript accordingly.

Please explain in detail for materials and methods 4.7, 4.8, 4.9, 4.11, 4.12.

Response: According to reviewer’s comments, we have described more details about the materials and methods and have also given the references as follows.

4.7 Lipid peroxidation assay

Cells (1.5 × 105 cells/mL) were cultured with eckol and/or PM2.5 for 24 h in the chamber slides. The lipid hydroperoxides in cells were reacted with diphenylpyrenylphosphine (DPPP; Molecular Probes) and the lipid adducts of DPPP oxide were detected by confocal microscope [30].

4.8 Protein carbonylation assay

Cells (1.5 × 105 cells/mL) were cultured with eckol and/or PM2.5 for 24 h in the culture dish. The OxiselectTM Protein Carbonyl ELISA kit (Cell Biolabs, San Diego, CA, USA) was used to detect protein oxidation following the manufacturer’s instructions [34].

4.9 Detection of 8-oxoguanine (8-oxoG)

Cells (1.5 × 105 cells/mL) were cultured with eckol and/or PM2.5 for 24 h in the chamber slides. The avidin-TRITC conjugate was used to detect 8-oxoG, an indicator of oxidative DNA damage. The stained cells were visualized under the confocal microscope [27].

4.11 Quantification of Ca2+ level

Cells (1.5 × 105 cells/mL) were cultured with eckol and/or PM2.5 for 24 h in the chamber slides. Cells were stained with Rhod-2 AM to check intracellular or mitochondrial calcium levels, respectively and the images were captured by confocal microscopy [30].

4.12 Mitochondrial membrane potential (ΔΨm) analysis

Cells (1.5 × 105 cells/ml) were cultured with eckol and/or PM2.5 for 24 h in the chamber slides. Cells were stained with 5,5′,6,6′-tetrachloro-1,1′,3,3′-tetraethylbenzimidazolylcarbocyanine iodide (JC-1, Invitrogen, Carlsbad, CA, USA) to observe changes in cell membrane potential. The fluorescence emission was analyzed by confocal microscopy and flow cytometry [6].

Reviewer 2 Report

The manuscript describes a study into a protective effect of eckol, extracted from brown algae, on oxidative stress, and associated phenomena in human keratinocytes in vitro.

The study provides a battery of complimentary parameters, and much effort had to be taken to conduct all of these analyses. 

Despite it, a manuscript offers a very brief introduction that fails to provide a full rationale, methodology which is too brief to be acceptable as it does not even fully elucidate experimental model, and discussion which should include elaboration on study limitations and future research prospects.

I provide a list of comments to each manuscript section, and encourage Authors to undertake a major revision of their manuscript. 

Abstract 

1. Provide an  eckol concentration 

2. I miss at least one sentence of conclusion   

 Introduction 

 1. Give Latin name of class ( Phaeophyceae) apart from its common English name.

 2. Manuscript would benefit from a figure of eckol structure

 3. When introducing the evidenced effects of eckol, provide a study model under which they were observed (in vitro, in vivo, clinical trial etc.), doses, route of application. This is essential information. 

4. What is meant by *modernization* in this context?

 5. I would suggest to highlight that air pollution, including PM emissions are a global health issue, and that number of reports address the effects on pulmonary system and to less extent, skin. 

6. Please include these review papers on the effect of air pollution on skin:  https://www.ncbi.nlm.nih.gov/m/pubmed/29186837/  https://www.mdpi.com/2079-9284/5/1/4  

7. You need to elaborate more on study aims: experimental model and studied parameters.

  8. Authors omit an important review on eckol which was published recently:  https://www.mdpi.com/1660-3397/17/6/361/htm  

 M&M 

Provide species name from which eckol was extracted.

 How was eckol extracted? Provide a detailed information on how was it obtained and purified from brown algae.

  How much eckol can be obtained from 1 g dw of brown algae?  I miss this information in the context of applied concentration.  

I suggest to provide *Experimental design* section and clearly define studied model, define eckol concentration and rationale for this concentration (why 30uM?), rationale for PM2.5 level, time and conditions of exposure, all studied parameters chosen for study,  number of independent repetitions, number of technical repetitions. In the subsequent sections define methodology for each parameter.  

What constituted a control?

Reference for each method should be given. Currently,  they are provided only for some. Moreover, methodology should be explained in detail. For some parameters a somewhat detailed info with rationale for assay is provided, for some other no details are given, and methodology is explained as a single sentence. Please use an unlimited space in journal to explain everything in detail. As a rule it is much better to provide as much methodology info as possible instead of providing too little.  

It would be better to avoid antibiotics in keratinocytes cultures ad these agents may affect their proliferation and other processes, see: https://www.ncbi.nlm.nih.gov/m/pubmed/26287649/

 Description of statistical analysis implies that data was normally distributed. Was it? If positive, state it along with method to test normality. If data differed from Gaussian distribution, re-analyse data using non-parametric methods, e.g. K-W Anova.  

Results

1. Since authors separated a discussion from results description, I would prefer to move rationale for each parameter in the last paragraph of Introduction in broader description of study aims.

2. Provide n in each figure caption. Moreover, provide eckol concentration and time of exposure for full clarity.

3. How long was the exposure in ROS assay? It is unclear from M&M nor Results.  

Discussion  

1. Study limitations should be outlined in the separate paragraph  

2. I miss discussion on future research prospects given the fact that this was only an in vitro study.  

3. Can you elaborate more on how eckol could practically be used in prevention of oxidative stress and its consequences for human skin? How much of it can you obtain from algae and can you make cosmetics that provide its sufficient levels? How could it be used in prevention of PM-induced damage? In other words how can you take in vitro results to a clinical level? 

Author Response

Reviewer comment 2.

The manuscript describes a study into a protective effect of eckol, extracted from brown algae, on oxidative stress, and associated phenomena in human keratinocytes in vitro. The study provides a battery of complimentary parameters, and much effort had to be taken to conduct all of these analyses. Despite it, a manuscript offers a very brief introduction that fails to provide a full rationale, methodology which is too brief to be acceptable as it does not even fully elucidate experimental model, and discussion which should include elaboration on study limitations and future research prospects. I provide a list of comments to each manuscript section, and encourage Authors to undertake a major revision of their manuscript. 

Abstract 

1. Provide an eckol concentration.

Response: According to reviewer’s comment, we have inserted the eckol concentration in the abstract as follows.

Furthermore, eckol (30 μM) decreased ROS generation, ensuring stability of molecules, and maintaining a steady mitochondrial state.

2. I miss at least one sentence of conclusion   

Response: We have included the conclusion after Materials and Methods section as follows, which summarizes the study results and suggests applications of eckol.

5. Conclusions

PM2.5 causes skin cell damage by generating ROS, excess of which oxidizes intracellular molecules and causes mitochondrial dysfunction, and activates MAPK signaling pathways. The skin is the first protection from various pollutants in the air, and it is important to identify effective biological compounds to prevent skin damage. Eckol has been known to inhibit UVB-induced ROS generation in keratinocytes [6]. In this study, we show that eckol could protect keratinocytes from PM2.5-induced apoptosis

by halting ROS generation, suggesting that eckol is a suitable candidate for skin protection and can be useful in the cosmetics industry as well as in medicine.

 Introduction 

1. Give Latin name of class (Phaeophyceae) apart from its common English name.

Response: According to reviewer’s comment, we have inserted the Latin name of class (Phaeophyceae) as follows.

”which is a kind of phlorotannin present in brown seaweeds (Phaeophyceae)”

2. Manuscript would benefit from a figure of eckol structure

Response: According to reviewer’s comment, we have inserted the chemical structure of eckol as Figure 1a.

3. When introducing the evidenced effects of eckol, provide a study model under which they were observed (in vitro, in vivo, clinical trial etc.), doses, route of application. This is essential information. 

Response: According to reviewer’s comment, we have provided more details in the introduction section. The newly added sentences are provided below.

Studies have shown that eckol, which is a kind of phlorotannin present in brown seaweeds (Phaeophyceae), decreases ultraviolet B (UVB) induced oxidative stress in human keratinocytes at a dose of 27 μM [6], and inhibits cancer in SKH-1 mice via inhibiting UVB-induced inflammation [7], and declines matrix metalloproteinase-1 level in human dermal fibroblasts implying anti-aging effects at a dose of 10 μM [8].

4. What is meant by *modernization* in this context?

Response: We realize that the term “modernization” can be subjective. Therefore, to avoid confusions, we have rewritten the sentences as shown below to remove the term “modernization”.

Air pollution by continuous emission of various pollutants into the atmosphere has led to rapid decline of public health and accelerated climate change [9]. In addition, more than 90% population in the world breathed unhealthy air in 2017, which suggests that particulate pollution is a global challenge [10].

5. I would suggest to highlight that air pollution, including PM emissions are a global health issue, and that number of reports address the effects on pulmonary system and to less extent, skin. 

Response: It is not suitable to write this sentence here. According to reviewer’s comment, we have corrected the sentence as Air pollution by continuous emission of various pollutants into the atmosphere has led to rapid decline of public health and accelerated climate change [9].”

6. I would suggest highlighting that air pollution, including PM emissions are a global health issue, and that number of reports address the effects on pulmonary system and to less extent, skin. Please include these review papers on the effect of air pollution on skin: https://www.ncbi.nlm.nih.gov/m/pubmed/29186837/ https://www.mdpi.com/2079-9284/5/1/4  

Response: According to reviewer’s suggestion, we have included the following sentence in the introduction section and cited a review paper (shown below).

“In addition, more than 90% population in the world breathed unhealthy air in 2017, which suggests that particulate pollution is a global challenge [10].”

Ngoc, L.; Park, D.; Lee, Y.; Lee Y.C. Systematic review and meta-analysis of human skin diseases due to particulate matter. Int. J. Environ. Res. Public Health 2017, 14, 1458.

7. You need to elaborate more on study aims: experimental model and studied parameters. Authors omit an important review on eckol which was published recently:  https://www.mdpi.com/1660-3397/17/6/361/htm  

Response: According to the reviewer’s suggestion, we have explained in the aim of the study in the Introduction section and inserted the recent reference well. The revised sentences are provided below.

“A recent study presents a comprehensive summary of the characteristics of eckol relevant for its therapeutic potential, including antioxidant activity following exposure to H2O2, radiation, and PM10 [17]. PM2.5 is a fine particulate matter that causes skin apoptosis related to the ROS generation [16], and it would be interesting to investigate whether Eckol protected skin cells from ROS-induced damage. Moreover, the mode of action of eckol on PM2.5 is not well-established. Here, we aimed to unravel the mechanism underlying the protective action of eckol on PM2.5-induced skin cell apoptosis.”

Manandhar, B.; Paudel P.; Seong S.H.; Jung, H.A.; Choi, J.S. Characterizing Eckol as a Therapeutic Aid: A Systematic Review. Mar. Drugs 2019, 17, E361.

M&M 

1. Provide species name from which eckol was extracted.

Response: According to the reviewer’s comment, we have included the details as shown below.

“Eckol (Figure 1a) was provided by Professor Nam Ho Lee of Jeju National University (Jeju, Republic of Korea), which belonged to Phaeophyceae.”

2. How was eckol extracted? Provide a detailed information on how was it obtained and purified from brown algae.

Response: According to the reviewer’s comment, we have included the details as shown below.

Preparation of the extract and its purification was following the reported protocol [44]. The dried brown alga Ecklonia Cava was extracted with 80% methanol and the crude extract was purified by HPLC.

Moon, C.; Kim, S.H.; Kim, J.C.; Hyun, J.W.; Lee, N.H.; Park, J.W.; Shin, T. Protective effect of phlorotannin components phloroglucinol and eckol on radiation-induced intestinal injury in mice. Phytother. Res. 200822, 238-242. 

3. How much eckol can be obtained from 1 g dw of brown algae?  I miss this information in the context of applied concentration.  

Response: According to the reviewer’s comment, we have provided more details about the extraction and purification of eckol.

“After purification, 20 mg of pure eckol was obtained from 1 kg dry weight of the brown algae.”

4. I suggest to provide *Experimental design* section and clearly define studied model, define eckol concentration and rationale for this concentration (why 30 uM?), rationale for PM2.5 level, time and conditions of exposure, all studied parameters chosen for study, number of independent repetitions, number of technical repetitions. In the subsequent sections define methodology for each parameter.

Response: According to the reviewer’s comment, in the Introduction section, we have provided supplementary explanation about the model used in the study. 

In the previous study, it was shown that eckol had no cytotoxicity against HaCaT keratinocytes up to a concentration of 50 μM; however, it showed antioxidant activity at 27 μM. The description is showed in the results part.

Kang, N.J.; Koo, D.H.; Kang. G.J.; Han, S.C.; Lee, B.W.; Koh, Y.S.; Hyun, J.W.; Lee, N.H.; Ko, M.H.; Kang, H.K.; Yoo, E.S. Dieckol, a Component of Ecklonia cava, Suppresses the Production of MDC/CCL22 via Down-Regulating STAT1 Pathway in Interferon-γ Stimulated HaCaT Human Keratinocytes. Biomol. Ther. (Seoul) 2015, 23, 238-244.

Later in the Discussion section, we have included the rationale for choosing the concentration eckol and PM2.5 level. Moreover, we have also included these parameters in each Figure.  The sentences included in the revised manuscript are as follows.

1) A recent study presents a comprehensive summary of the characteristics of eckol relevant for its therapeutic potential, including antioxidant activity following exposure to H2O2, radiation, and PM10 [17]. PM2.5 is a fine particulate matter that causes skin apoptosis related to the ROS generation [16], and it would be interesting to investigate whether Eckol protected skin cells from ROS-induced damage. Moreover, the mode of action of eckol on PM2.5 is not well-established. Based on this hypothesis, we were interested to unravel the mechanism….

2) In our earlier studies (Piao et al., 2018), we conducted experiments with different concentrations of PM2.5 at 25, 50, 75 and 100 µg/mL for 24 hours in cells. In this study, we found that a concentrations of 75 µg/mL PM2.5 caused necrosis in keratinocytes. For the research on oxidative stress, we decided to use PM2.5 (50 µg/mL). This rationale is included in the manuscript as shown below. “Piao et al studied PM2.5-induced ROS generation at different concentrations (25, 50, 75, and 100 µg/mL) for 24 h in HaCaT keratinocytes, and found that PM2.5 50 µg/mL caused excessive ROS and skin dysfunction [29].

3) All experiments were performed after treatment with PM2.5 for 24h, and n=3 for every group.

5. What constituted a control?

Response: We used only DMSO solution (without eckol) as control for our experiment. This is now mentioned in the Materials and Methods section. “DMSO was as the control.”

6. Reference for each method should be given. Currently, they are provided only for some. Moreover, methodology should be explained in detail. For some parameters a somewhat detailed info with rationale for assay is provided, for some other no details are given, and methodology is explained as a single sentence. Please use an unlimited space in journal to explain everything in detail. As a rule it is much better to provide as much methodology info as possible instead of providing too little.  

Response: According to the reviewer’s comment, we have revised our Materials and Methods section to include all references.

4.7 Lipid peroxidation assay

Cells (1.5 × 105 cells/mL) were cultured with eckol and/or PM2.5 for 24 h in the chamber slides. The lipid hydroperoxides in cells were reacted with diphenylpyrenylphosphine (DPPP; Molecular Probes) and the lipid adducts of DPPP oxide were detected by confocal microscope [30].

4.8 Protein carbonylation assay

        Cells (1.5 × 105 cells/mL) were cultured with eckol and/or PM2.5 for 24 h in the culture dish. The OxiselectTM Protein Carbonyl ELISA kit (Cell Biolabs, San Diego, CA, USA) was used to detect protein oxidation following the manufacturer’s instructions [34].

4.9 Detection of 8-oxoguanine (8-oxoG)

Cells (1.5 × 105 cells/mL) were cultured with eckol and/or PM2.5 for 24 h in the chamber slides. The avidin-TRITC conjugate was used to detect 8-oxoG, an indicator of oxidative DNA damage. The stained cells were visualized under the confocal microscope [27].

4.10 Single cell gel electrophoresis (Comet assay)

Cells (1.5 × 105 cells/mL) were cultured with eckol and/or PM2.5 for 30 min in the microtubes, and the harvested cells were fixed on a microscopic slide with low-melting agarose (1%). The slides with cells were permeated into lysis buffer (pH 10) for 1 h, which contained NaCl (2.5 M), Na-EDTA (100 mM), Tris (10 mM), Triton X-100 (1%), and DMSO (10%). Finally, the slides were subjected to electrophoresis, and the samples were dyed with ethidium bromide. The fluorescent images of the tails were captured with a fluorescence microscope and the tail lengths (50 cells per slide) were analyzed by the image analysis software (Kinetic Imaging, Komet 5.5, Liverpool, UK) [29].

4.11 Quantification of Ca2+ level

Cells (1.5 × 105 cells/mL) were cultured with eckol and/or PM2.5 for 24 h in the chamber slides. Cells were stained with Rhod-2 AM to check intracellular or mitochondrial calcium levels, respectively and the images were captured by confocal microscopy [30].

4.12 Mitochondrial membrane potential (ΔΨm) analysis

Cells (1.5 × 105 cells/mL) were cultured with eckol and/or PM2.5 for 24 h in the chamber slides. Cells were stained with 5,5′,6,6′-tetrachloro-1,1′,3,3′-tetraethylbenzimidazolylcarbocyanine iodide (JC-1, Invitrogen, Carlsbad, CA, USA) to observe changes in cell membrane potential. The fluorescence emission was analyzed by confocal microscopy and flow cytometry [6].

7. It would be better to avoid antibiotics in keratinocytes cultures at these agents may affect their proliferation and other processes, see: https://www.ncbi.nlm.nih.gov/m/pubmed/26287649/

Response: We agree with the reviewer’s concern about the effect of antibiotics in keratinocytes culture. However, antibiotics could kill all cells, which could lead to more serious consequences. Moreover, it has been shown that in standard tissue culture, antimicrobial agents penicillin (10,000 units/ml)-streptomycin (10,000 micrograms/ml)-amphotericin B (25 micrograms/ml) have no effect on the growth rates of keratinocyte.  Notably, the concentration of the antibiotic used was about 10 times higher than what we had used in this study.  Following is the reference. 

Cooper ML, Boyce ST, Hansbrough JF, Foreman TJ, Frank DH. Cytotoxicity to cultured human keratinocytes of topical antimicrobial agents. J Surg Res. 1990;48(3):190-5.

However, in the reference cited by us and the one cited by the reviewer, the cells used were NHK and not HaCaT keratinocyte. Therefore, these two reports may not truly represent the real conditions in our experiment.

8. Description of statistical analysis implies that data was normally distributed. Was it? If positive, state it along with method to test normality. If data differed from Gaussian distribution, re-analyse data using non-parametric methods, e.g. K-W Anova.  

Response: We have given more details about the statistical analysis as follows.     All data were collected from three experiments and presented as mean ± standard error, which were analyzed by variance (ANOVA) with Tukey’s test using Sigma Stat (v12) software (SPSS, Chicago, IL, USA).

Results

1. Since authors separated a discussion from results description, I would prefer to move rationale for each parameter in the last paragraph of Introduction in broader description of study aims.

Response: According to the reviewer’s comment, we have explained the study aims in the Introduction section and inserted the reference as follows.

Here, we have investigated the potential benefits of eckol on keratinocytes by studying its inhibitory effect on molecular damage, mitochondrial dysfunction, apoptosis related factors, and MAPK signaling related proteins. In this study, our aim was also to gain insights into the mechanism underlying the protective action of eckol on PM2.5-induced skin cell apoptosis.

2. Provide n in each figure caption. Moreover, provide eckol concentration and time of exposure for full clarity.

Response: According to the reviewer’s comment, we have provided the experimental details in all the Figures. 

All experiments were performed after treatment with PM2.5 for 24 h, and n=3 for every group.

3. How long was the exposure in ROS assay? It is unclear from M&M nor Results.

Response: According to the reviewer’s comment, we have included the exposure time in the description of ROS assay as follows.

Cells (1.5 × 105 cells/mL) were treated with eckol (30 µM) for 30 min, PM2.5 (50 µg/mL) for another 24 h, and DCF-DA (25 µM) sequentially.

Discussion  

1. Study limitations should be outlined in the separate paragraph. I miss discussion on future research prospects given the fact that this was only an in vitro study.

Response: According to the reviewer’s comment, we have revised the manuscript to include a paragraph on limitations and future research prospects in the Discussion section.

Although the protective effects of eckol on human keratinocytes from PM2.5-induced skin damage has been shown, there are limitation to this study. These results from in vitro experiments need to be validated by animal studies and clinical trials. Moreover, the concentration of air pollutants in the natural environment is different from the PM2.5 purchased from the company, which provide certain ingredients for reference. In the future, there should be in vivo animal trials on skin protection to elucidate the protective effects and side effects of eckol under the complicated living environments.

2. Can you elaborate more on how eckol could practically be used in prevention of oxidative stress and its consequences for human skin? How much of it can you obtain from algae and can you make cosmetics that provide its sufficient levels? How could it be used in prevention of PM-induced damage? In other words how can you take in vitro results to a clinical level? 

Response: In this in vitro study, we demonstrated the protective effects of eckol on keratinocytes. While this points to potential therapeutic and cosmetic applications, it remains to be validated by in vivo studies and in human clinical studies. These experiments will be performed in the future.

Reviewer 3 Report

In the revised version of the paper entitled:"Eckol Inhibits Particulate Matter 2.5-induced Skin Keratinocyte Damage via MAPK Signaling Pathway", the Authors reported an in vitro study testing the mechanism underlying the protective effects of eckol, a phlorotannin isolated from brown sea-weed, towards human HaCaT keratinocytes from particulate matter 2.5-induced cell damage. The paper is interesting and scientifically sound; moreover, in the revised version, the Authors addressed carefully most of the points raised by the Reviewers.

Just the following few minor points should be checked:

- As stated by Reviewer 2, it shoould be helpful to include a sentence of conclusion in the abstract

- In the result section, I suggest to add the p values even within the text. Moreover, I suggest to delete the sentences at the end of each paragraph of the results; that sentences (e.g., at the end of paragraph 2.1:"Therefore, eckol could protect keratinocytes 8 from PM2.5-induced apoptotic cell death by inhibiting ROS generation") sound more as a comment than a result, and it would be preferable that in the Results section mostly facts are reported.

Author Response

Reviewer comment 3.

In the revised version of the paper entitled:"Eckol Inhibits Particulate Matter 2.5-induced Skin Keratinocyte Damage via MAPK Signaling Pathway", the Authors reported an in vitro study testing the mechanism underlying the protective effects of eckol, a phlorotannin isolated from brown sea-weed, towards human HaCaT keratinocytes from particulate matter 2.5-induced cell damage. The paper is interesting and scientifically sound; moreover, in the revised version, the Authors addressed carefully most of the points raised by the Reviewers.

Just the following few minor points should be checked:

- As stated by Reviewer 2, it should be helpful to include a sentence of conclusion in the abstract.

Response According to reviewer’s comments, we have inserted the conclusion in the abstract as follows; In conclusion, eckol could protect skin HaCaT cells from PM2.5 induced apoptosis via inhibiting ROS generation.

- In the result section, I suggest to add the p values even within the text. Moreover, I suggest to delete the sentences at the end of each paragraph of the results; that sentences (e.g., at the end of paragraph 2.1:"Therefore, eckol could protect keratinocytes 8 from PM2.5-induced apoptotic cell death by inhibiting ROS generation") sound more as a comment than a result, and it would be preferable that in the Results section mostly facts are reported.

Response According to reviewer’s comments, we have deleted those sentences.

Round 2

Reviewer 2 Report

I appreciate all revisions made to this manuscript. I think it looks much better and is more complete. I only have some minor corrections that I have placed directly on manuscript file.

Author Response

Reviewer comment 2.

I appreciate all revisions made to this manuscript. I think it looks much better and is more complete. I only have some minor corrections that I have placed directly on manuscript file.

Response Thank you for your contribution to this paper.